# Analysis of the Forecast Price as a Factor of Sustainable Development of Agriculture

**Maxim Tatarintsev** [1,†][ID]**, Sergey Korchagin** [1,†][ID]**, Petr Nikitin** [1,†][ID]**, Rimma Gorokhova** [1,*,†][ID]**, Irina Bystrenina** [2,†][ID] **and Denis Serdechnyy** [3,†][ID]

1 Department of Data Analysis and Machine Learning, Financial University under the Government of the Russian Federation, 38 Shcherbakovskaya, 105187 Moscow, Russia; maks.tatarintsev@inbox.ru (M.T.); sakorchagin@fa.ru (S.K.); pvnikitin@fa.ru (P.N.)
2 Department of Applied Informatics, Russian State Agrarian University—Moscow Timiryazev Agricultural Academy, 49 Timiryazevskaya, 127550 Moscow, Russia; iesh@rambler.ru
3 Department of Innovation Management, State University of Management, 99 prosp. Rjazanskij, 109542 Moscow, Russia; Dv_serdechnyj@guu.ru
* Correspondence: rigorokhova@fa.ru
† These authors contributed equally to this work.

**Abstract:** Analysis of the rise in prices for consumer goods is a state's priority task. The state assumes the obligation to regulate pricing in all spheres of consumption. First of all, the prices for essential commodities to which agricultural products belong are analyzed. The article shows the changes in prices for consumer goods of agricultural products (sugar) during a pandemic. The analysis of forecasting prices for sugar and its impact on the development of its production is carried out. The construction of the forecast model was based on extrapolation. The structure of a forecast model for price changes was based on the analysis of the time series of the Autoregressive Integrated Moving Average (ARIMA) class. This model consists of an autoregressive model and a moving average model. A forecast of the volume of domestic sugar transportation by rail has been completed. The algorithms implemented this model for searching for initial approximations and optimal parameters for the predictive model. The Hirotsugu Akaike Information Criterion (AIC) was used to select the best model. The algorithms were implemented in the Python programming language. The quality check of the description was performed with a predictive model of actual data. An economic interpretation of the rise in sugar prices and proof of the forecast's truth obtained from a financial point of view were carried out.

**Keywords:** agronomy; machine learning; predictive analytics; autoregressive integrated moving average; Box–Cox transform



## 1. Introduction

### 1.1. Problem Statement

Agriculture is a large and multi-stage industry. The sustainability of the development of this industry is the primary task of the state. Therefore, in today's circumstances, it is possible to exercise qualitative control over each sector only when designing, using and implementing effective forecasting and planning mechanisms. A comprehensive assessment of the state of the future makes it possible to study promising development programs and evaluate the consequences of decisions taken at a given time. The coronavirus pandemic has caused severe damage to all major world powers' economies. Russia is no exception: high mortality, global instability, job losses, and declining wages— no analyst forecast predicted such a large-scale disaster. In 2020, there has been a constant rise in goods' prices. According to Rosstat estimates, fruits and vegetables have risen in price by 17.40% sunflower oil by 25.91%, and cereals by 20.12% for this period. However, the most significant increase in consumer prices fell on sugar— 64.54%. Due to such negative dynamics, it was decided to analyze this area.

In Russian practice, the primary method for the production of granulated sugar is the processing of sugar beets. Consequently, a seasonal component appears [1,2]. The amount of vegetables harvested is influenced by the area planted and the yield. The more space is planted with fertile vegetables, the more abundant the harvest will be. Next, the harvested beets are transported to unique factories, where the root crop will be processed into sugar. The next stage is the transportation of products from the manufacturer to the consumer. In the domestic market, rail and freight delivery to the point of sale is carried out. Thus, transportation is a critical link in the chain described above. The target variable for our study is the price of sugar. The forecast of the sugar price will be carried out, taking into account the volume of sugar transportation in the country (using the example of the Russian Federation) from 2020 to 2023. Product transport data from 1 January 2011 to 31 December 2019 will be selected as input data for the simulation. In this case, the base time interval in the analysis will be considered the calendar month of each year and the corresponding indicator characterizing the amount of sugar transported during this period. Having built a predictive model and ensuring its accuracy, we will determine the trend towards decreasing or increasing the volume of domestic traffic of the product in question. Based on this result, we will predict the dynamics of sugar prices in the short term. For this, we consider the inverse correlation of the demand for a product with the price itself—an increase in value will guarantee a decrease in order. In this case, the volume of cargo transportation in the country will also decrease. Since we will already know the amount of sugar transported over the following years, we will estimate the cost measure of the product itself [3].

*1.2. Justification of the Choice of Methods for Solving the Problem*

There are two broad approaches to forecasting: extrapolation (the models are based on the experience) and modeling (the study of the dependence of factors affecting a parameter is used as a basis). The choice must be made based on both the study and the available data. The research will be based on data from previous years on the volume of transported sugar in the country for different time intervals. In the analysis, we will rely on a time series. As a forecasting method, we will choose extrapolation since the internal transportation of sugar is characterized by the time and volume of the transported cargo. Many models allow forecasting with varying degrees of accuracy: correlation–regression analysis [4–7], neural network models [8,9], research-based on multiple regression [10–12], models based on classification–regression trees [13,14], maximum likelihood sampling models [15–17] and many others [18–21]. For forecasting economic time series, models of the ARIMA class are used [22]. Our study will use the ARIMA and Seasonal Autoregressive Integrated Moving Average (SARIMA) models as there is a seasonal component in the data. The relevance of the application of these methods is confirmed in some studies by leading scientists. Some studies were modeled and predicted using ARIMA, SARIMA, and other stochastic models and related to agricultural productivity: water resources, hydrology and quality, and others [23–26].

## 2. Data Preparation and Transformation
*2.1. Time Series: Visualization and Checking for Stationarity*

Statistically reliable data were taken from the Rosstat estimate for the selected period. We have chosen a rather long observation period—from 2010 to 2018. In total, 100 observations were considered in the study. Let us put all the data on a graph (Figure 1) and conduct a visual analysis. Any time series can theoretically be decomposed into several key components: trend, seasonal component, cyclical component, and some random component. There is a pronounced annual seasonality, that is, the second half of each year has maximum values.

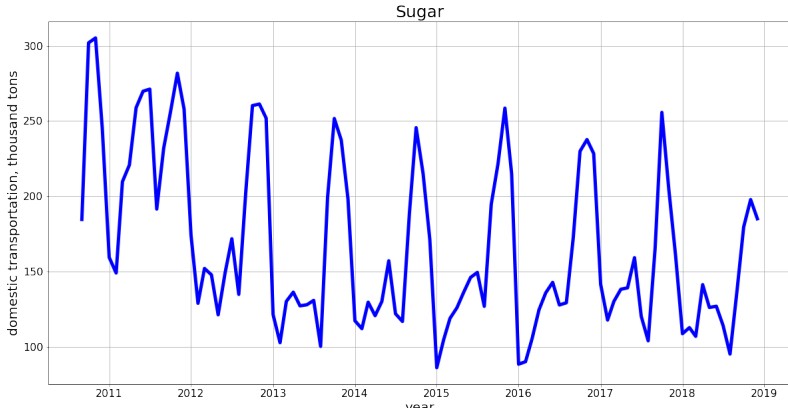

**Figure 1.** Domestic rail transport of sugar in the Russian Federation.

Let us check our time series for stationarity. Unfortunately, there is no one-size-fits-all method. Each time series is exceptional, and its study algorithm must be selected individually. Yang L., Lee C., Su J.-J. used the Dickey–Fuller test in their studies [27]. This work also used the Dickey–Fuller statistical test since it has a significant test power for series with many observations. The assumption that the series is nonstationary is taken as the null hypothesis. If during the experiment it turns out that the *p*-value of the time series is less than 0.05, that is, the 95.00% significance level is reached, then there will be grounds to reject the null hypothesis. Using the Python toolkit, this can be done as follows (Figure 2):

```
1  a=sm.tsa.stattools.adfuller(sugar.tons)
2  if sm.tsa.stattools.adfuller(sugar.tons)[1] <0.05:
3      print("p-value= ", round(sm.tsa.stattools.adfuller(sugar.tons)[1],4))
4  else:
5      print("p-value= ", round(sm.tsa.stattools.adfuller(sugar.tons)[1],4))

p-value=  0.0104
```

**Figure 2.** Implementation of the Dickey–Fuller test in Python.

The Dickey–Fuller criterion gives an attainable significance level of 0.010. Consequently, the null hypothesis is rejected, so the series is considered stationary. However, the graph shows that the trend is visible. Since we could not conclude the presence or absence of a trend in the series, we will apply another very effective criterion of inversions. It is advisable first to formalize the hypotheses:

$H_0 : E_i = x, i = 1, 2, 3, \ldots, N-$ no trend

$H_1 : IE_{i+1} + E_i I > 0, i = 1, 2, 3, \ldots, N-1-$ trend

Next, it is necessary to calculate how many times an inequality of the form $x_i > x_j$, $i < j$. Let us denote this quantity as $I$. The area of acceptance of the null hypothesis is as follows (1):

$$I_{100;1-\alpha/2} < I < I_{100;\alpha/2} \tag{1}$$

Now, let us search for the normalized statistics $I\_$ and the number of inversions in our time series using the inversion criterion (Figure 3).

According to the corresponding criterion table at a significance level of 0.05, we can find:

$I_{100;1-\alpha/2} = I_{100;0.975} = 2145$

$I_{100;\alpha/2} = I_{100;0.025} = 2751$

Therefore, hypothesis $H_0$ should be rejected with a 5% significance level since $I = 3086$ does not fall within the confidence interval between 2145 and 2751, resulting in a non-stationary time series. It is advisable to check this statement using the Seasonal-Trend Decomposition Procedure Based on Loess (Figure 4).

```
1   n=len(sugar.tons)
2   #Looking for inversions for each element
3   index=0
4   inversion1=0
5   zero=0
6   list1=[]
7   for i in sugar.tons:
8       for i in sugar.tons[zero:]:
9           if sugar.tons[zero]>sugar.tons[index]:
10              inversion1+=1
11          index+=1
12      list1.append(inversion1)
13      zero+=1
14      index=zero
15      inversion1=0
16
17  list1.pop(-1)
18  I=sum(list1)#total number of inversions in the sample
19  I_=(I-n*(n-1)/4)/((2*n**3+3*n**2-5*n)/72)**(1/2)
20  I_#normalized statistics that are described by the standard normal distribution law
21  I
```

3086

**Figure 3.** Implementation of the statistical test of inversions in Python.

```
1   rcParams['figure.figsize'] = 15, 8
2   decomposition = sm.tsa.seasonal_decompose(sugar.tons, model='additive',freq=12, extrapolate_trend = 12)
3   fig = decomposition.plot()
4   plt.show()
```

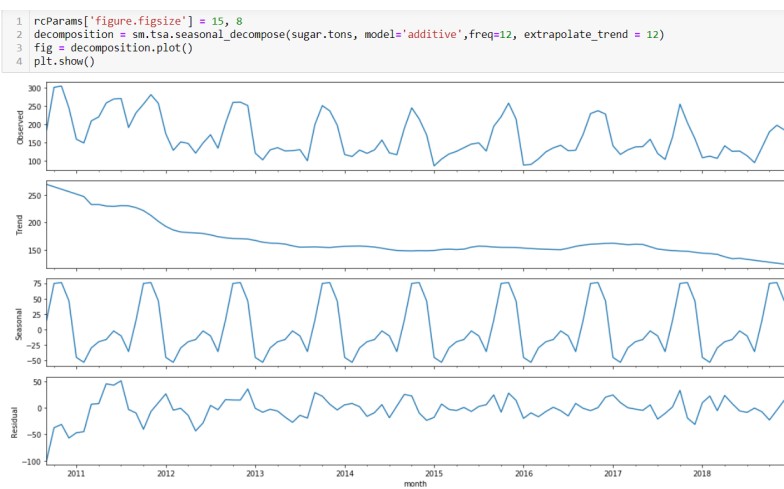

**Figure 4.** Seasonal-Trend Decomposition Procedure Based on Loess.

In Figure 4, the trend shows the general downward direction of the data. Consequently, the series is unambiguously nonstationary. Information from graphs 3 and 4 will be of great value in further research. The seasonal and residual information will be critical in further developing the forecast model. This conclusion is confirmed by the studies carried out in this direction by Guisande C., Rueda-Quecho A.J., Rangel-Silva F.A., Ríos-Vasquez J.M., Xiong T., Li C., Bao Y., Nguyen L., Novák V. [28–30].

### 2.2. Box–Cox Transform and Data Stabilization

In most cases, the data in the series have a noise component. This component affects the correctness of the assessment and the adequacy of the conclusions. It is possible to facilitate the task and level the arising noise components by smoothing the time series. This operation makes the data more stable and more convenient to work with. One of the most effective is the Box–Cox transformation. The application of this method can be noted in a number of studies by Voyant C., Notton G., Duchaud J.-L., Almorox J., Yaseen Z.M., De Lima e Silva P.C., Severiano C.A., Alves M.A., Silva R., Cohen M.W., Guimarães F.G. [31,32]. In a situation where $\lambda$ tends to zero, the variable will result in a logarithmic form, while at $\lambda = 1$—to linear. By varying $\lambda$, you can constantly change from linear to logarithmic and vice versa. The model will be smoother and more stable, which will lead to its good quality. We convert the series and find the optimal parameter $\lambda$ (Figure 5):

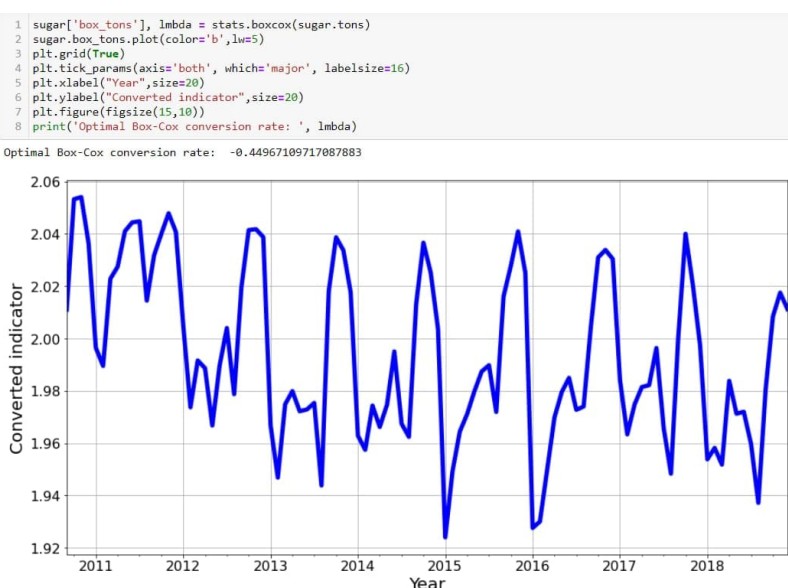

```
1  sugar['box_tons'], lmbda = stats.boxcox(sugar.tons)
2  sugar.box_tons.plot(color='b',lw=5)
3  plt.grid(True)
4  plt.tick_params(axis='both', which='major', labelsize=16)
5  plt.xlabel("Year",size=20)
6  plt.ylabel("Converted indicator",size=20)
7  plt.figure(figsize(15,10))
8  print('Optimal Box-Cox conversion rate: ', lmbda)
```

Optimal Box-Cox conversion rate:  -0.44967109717087883

**Figure 5.** Time series transformation using the Box–Cox test.

The graph shows that the program has found a suitable parameter $\lambda = -0.449$. The chart shows that the values on the ordinate have stabilized and are concentrated around a constant. Our new series may now be stationary. This can be checked using the previously used Dickey–Fuller criterion (Figure 6):

```
1  print('The Dickey-Fuller criterion:' , sm.tsa.stattools.adfuller(sugar.box_tons)[1])
```

The Dickey-Fuller criterion: 0.04356104005639127

**Figure 6.** Applying the Dickey–Fuller test to the transformed series.

The $p$-value turned out to be 0.044. That is, the series can be considered stationary. However, visually, we see again that there is a decreasing nonlinear trend. We conclude that the series, despite the applicable criterion's results, cannot be called stationary [33]. When analyzing data, you cannot rely entirely on the fairness of standards or methods, as they do not always give the expected results.

*2.3. Time Series Differentiation*

Some techniques allow you to transform the original series into a stationary one. The most effective and easy to use is the method of differentiation, which enables you to stabilize the average value of the series and remove various changes in its level. The principle of the operation of such a method is to subtract the previous observation from the current one. As a result, we will obtain a calculated "series of differences", which can be further investigated. Differentiation is seasonal and common. They can be applied repeatedly both separately from each other and in combination. For example, if a series has a pronounced seasonality, then it is always recommended to conduct seasonal differentiation first. In some cases, the series may be stationary after the first iteration. We apply the above procedure to our series (Figure 7):

Considering the optimal parameter of the Dickey–Fuller criterion [34], we conclude that the series is stationary. Let us carry out one more normal differentiation to consolidate this result (Figure 8). The series is now unambiguously stationary. The trend type has changed its structure. If there was a smoothly descending trend in the original time series, there is no systematic behavior—the values arbitrarily fluctuate around a constant. This is precisely the result we wanted to achieve by differentiation. The transformed time series is stationary. Now, we can build a predictive model.

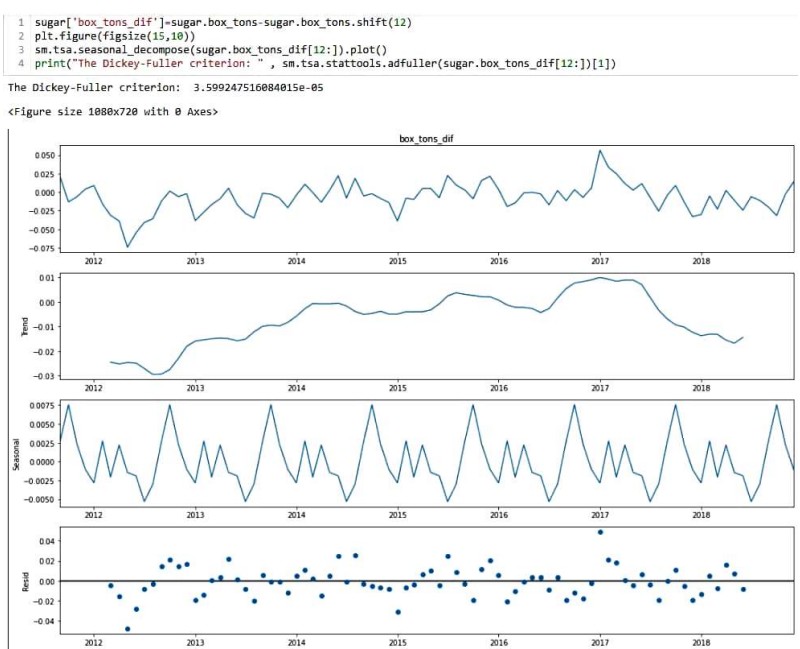

**Figure 7.** Application of seasonal differentiation and decomposition of a time series into components.

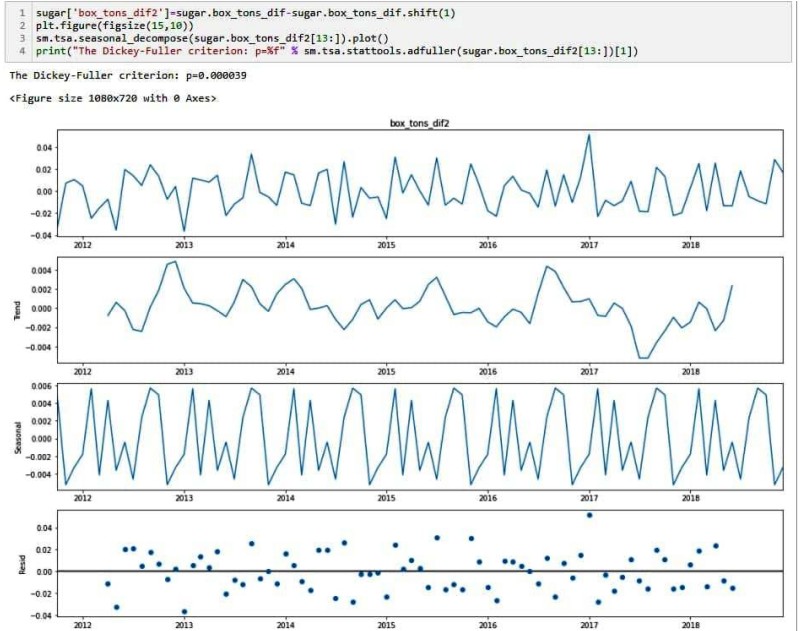

**Figure 8.** Application of normal differentiation and decomposition of a time series into components.

## 3. Making a Forecast of the Volume of Domestic Sugar Shipments by Rail

### 3.1. Finding Initial Approximations for the Predictive Model

Consider the autocorrelation [35] and partial autocorrelation functions [36] for the resulting transformed time series. In a general sense, an autocorrelation function can be understood as a measure of the linearity of the relationship between elements of a time series that are distant from each other by a certain number of points in time. It is possible to quantify the degree of similarity between the values of the series at neighboring points with the help of such a function. That is, it is the usual Pearson correlation coefficient [37] between the values of a given time series and its copy shifted by a certain number of values. The partial autocorrelation function is the autocorrelation of the residuals of the previous order autoregressive. It allows you to see the periodicity in the data and find the autoregressive order of the time series. With the help of graphs, we will look for

parameters for our model. We visualize with the use of Python two correlograms for the two time-differentiated series obtained in the previous paragraph (Figure 9):

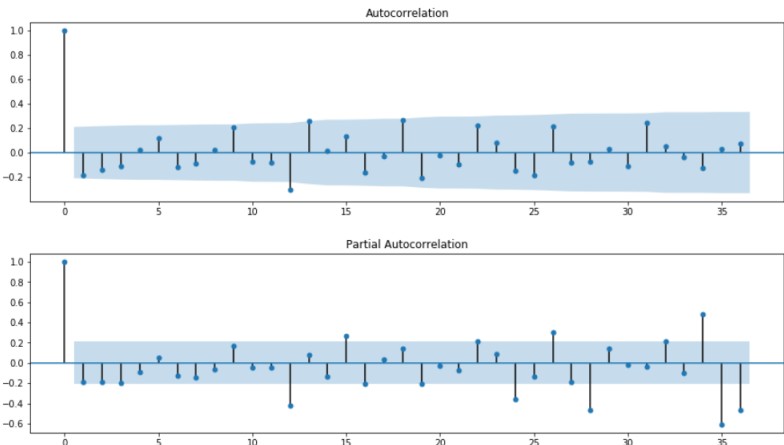

**Figure 9.** Autocorrelation and partial autocorrelation function graph.

Let us limit the number of lags along the abscissa axis to 36 so that the graph is not noisy. The blue corridor, which predominantly contains points, is the confidence interval (automatically set at 95%). Values outside this range are likely to be correlated rather than statistical randomness. So, let us proceed to select initial approximations for the forecast model [38] Let us first consider the graph of the autocorrelation function. To determine the initial approximation for the Q parameter, you need to pay attention to the maximum seasonal lag, which differs from 0. The seasonality period in our time series is one year or 12 months. We must find the largest value, which will be a multiple of 12. At the same time, it must necessarily go beyond the blue confidence corridor. The correlogram shows that the first and only value that satisfies us is 12. Therefore, determine that $Q = 1$. From one graph of Figure 9, you can also identify $q$. In our case, 12, 13 and 18 lags significantly differ from 0. It is not possible to take the seasonal coefficient as an initial approximation. Briefly, 18 lag is too large and will not be used in further work. There is only one number left, which will represent $q - 13$. Next, we turn to the plot of partial autocorrelation to find initial approximations for the parameters $P$ and $p$. The logic here will be identical to that used in the first correlogram analysis. So, the graph shows that $P$ and $p$ are equal to 3.

### 3.2. Finding the Optimal Parameters for the Predictive Model

The resulting SARIMA model consists of the following key components: $P = p = 3$, $Q = 1, q = 13$. Let us take $d$ and $D$ as the number of times the series' regular and seasonal differentiation is carried out, respectively. Since we performed both actions once, the parameters will be equal to 1. Consideration of the vector representations of the values shows that there will be 448 possible sets. From these combinations, it is necessary to choose the optimal one, that is, to set up the forecast model. The AIC allows you to choose the best model from several proposed. With the correct selection of initial approximations, the criterion will choose an optimal set of parameters from the set of values. We assume that several of the 448 models will be outliers. When hitting them, the row will diverge, the program will issue an error notification, thereby interrupting the iteration cycle. We will ignore every unsuitable parameter and consider only those that are similar to optimal. From the rest, those will be selected for which the value of the information criterion will be minimal. The Python implementation is shown in Figure 10:

```
1   %%time
2   results=[]
3   best_aic = float("inf")
4   warnings.filterwarnings('ignore')
5
6   for par in params_list:
7       try:
8           model=sm.tsa.statespace.SARIMAX(sugar.box_tons, order=(par[0],d,par[1]),\
9                           seasonal_order=(par[2], D, par[3], 12)).fit(disp=-1)
10      except:
11          print ('wrong parameters:',par)
12          continue
13      aic=model.aic
14      #print(aic)
15      if aic<best_aic:
16          best_model=model
17          best_aic=aic
18          best_par=par
19      results.append([par,model.aic])
20  warnings.filterwarnings('default')
```

**Figure 10.** Application of the Akaike information criterion.

We ended up with 32 unsuitable models. The quality of the selection of parameters is burdened by the program execution time—with sufficiently large initial approximations, it can be large (compilation took 15 minutes). So, the best were the sets $(1, 1, 1, 1)$, $(1, 1, 0, 1)$, $(2, 1, 0, 1)$, $(1, 1, 2, 0)$, with the values of the information criterion $-478.70$, $-478.60$, $-476.60$, $-475.80$, respectively. Since it is necessary to select the minimum values, we select the first combination. As a result, we got the SARIMA model $(1, 1, 1) * (1, 1, 1, 12)$. Simultaneously, the achieved level of significance $Q$ is high $-0.65$ (Figure 11).

```
==================================================================================
Ljung-Box (L1) (Q):                0.21   Jarque-Bera (JB):              0.55
Prob(Q):                           0.65   Prob(JB):                      0.76
Heteroskedasticity (H):            0.84   Skew:                         -0.14
Prob(H) (two-sided):               0.64   Kurtosis:                      3.28
==================================================================================
```

**Figure 11.** Statistical indicators of the model.

### 3.3. Analysis of the quality of the constructed model

Let us analyze the residuals. The residuals can be called the difference between actual data and predicted values. By this critical component of the time series, the quality of the constructed model is judged. In addition, using the residuals, one can say about the presence of any fundamental or not very gross errors [39]. Visually, according to the correlogram (Figure 12), it is possible to determine spontaneous behavior.

**Figure 12.** Results of the application of the Dickey–Fuller test and the Student's test. Building a correlogram.

There is no seasonality or trend here. We can visually see that the remnants look like noise. However, visual analysis is not enough to draw definitive conclusions. Therefore, it is necessary to show that the model works stably using a theoretical approach. There are several essential properties inherent in adequate residuals—unbiasedness, non-autocorrelation, stationarity [40]. Let us check each item. The first thing to check is unbiasedness. This term is understood as the equality of the mean value to zero. To do this, we will use the Student's test. The achieved significance level is approximately 0.720, which means that the hypothesis of unbiasedness is not rejected. The Dickey–Fuller criterion quite unambiguously confirms that the idea of non-stationarity is denied. That is, the remainders are stationary. The absence of autocorrelation can be justified by considering the correlogram in the previous paragraph. The high attainable level of significance confirms this. As a result, the residuals are unbiased, stationary, and non-autocorrelated. The fulfillment of all the necessary properties guarantees good trainability of the predictive model.

*3.4. Building a Predictive Model*

Let us check how well our predictive model will describe the initial data (Figure 13).

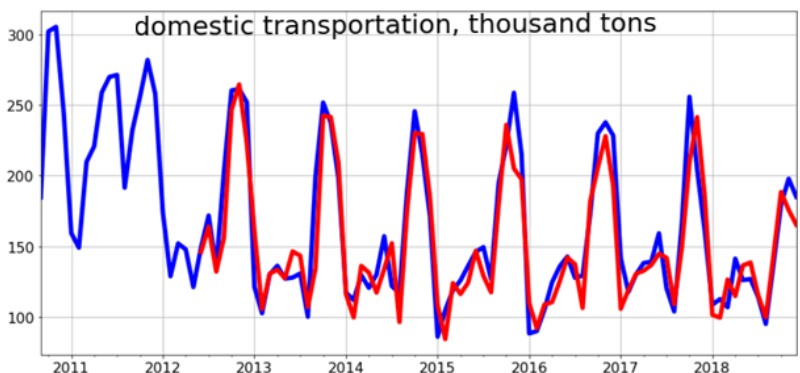

**Figure 13.** Checking the quality of the description of the predictive model of actual data.

The predicted values of the model's results are highlighted in red on the chart. Visual analysis reveals good data fit. This result can be confirmed by searching for the mean squared difference (MSE) between estimated and actual values (Figure 14). The obtained result— 1.105 shows that under the conditions of the study, it can be considered a permissible deviation. The high level of similarity between the present and predicted values is indicated by the high correlation coefficient— 0.920.

```
1  from sklearn.metrics import mean_squared_error
2  from scipy.stats import pearsonr
3  cor=pearsonr(predict,real)
4  predict = np.array(sugar.model[21:])
5  real = np.array(sugar.tons[21:])
6  MSE = mean_squared_error(real,predict)
7  print('MSE=',MSE)
8  print('Correlation coefficient=',cor[0])
```

```
MSE= 1.1046026028758928
Correlation coefficient= 0.9184981466005884
```

**Figure 14.** Finding the mean squared difference (MSE) between estimated and real values.

Let us apply the inverse Box–Cox transformation to return our changed series to its original form. Let us carry out the forecast for four years ahead (Figure 15).

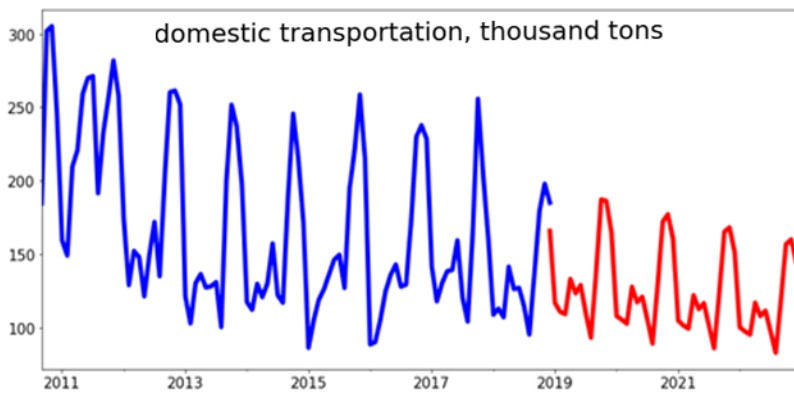

**Figure 15.** Building a forecast.

The trained model claims that domestic rail transport of sugar declined in 2020. In subsequent years, the volume of deliveries will remain at the level of the date in question. Since the price of sugar is inversely correlated with domestic supply, the cost measure should rise.

## 4. Economic Interpretation of Rising Sugar Prices

Let us prove our forecast's truth from an economic point of view. The trend towards reducing the volume of domestic transport of sugar by rail is apparent. This may be due to many different applied factors, including the importance of imports and exports of products, logistic redistribution of transport load, changes in the agricultural sector, the ratio of supply and demand for goods. We will analyze each of the listed segments, identify patterns and compare them with the forecast results to make a competent conclusion. The first step is to build graphs for the import (Figure 16) and export (Figure 17) of sugar.

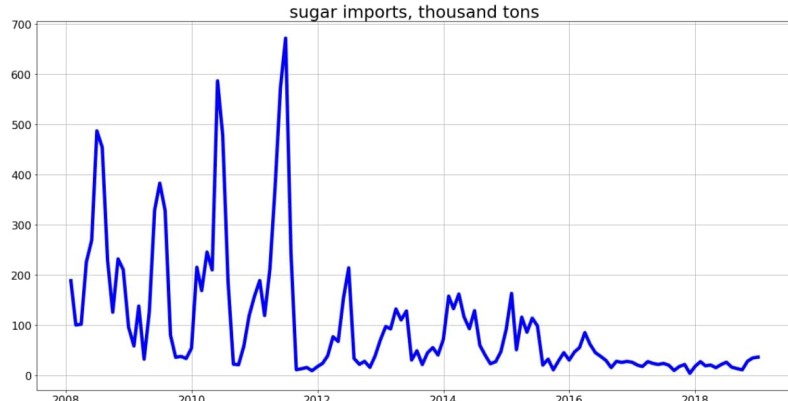

**Figure 16.** Import of sugar from Russia.

There has been a gradual decrease in the volume of sugar supplies from abroad over the past ten years. However, this indicator is still far from 0.

The reverse dynamics can be seen in the supply of goods abroad. The export level is actively increasing from 2016 to 2019. A contradiction arises: with a sufficient number of products in the country for self-sufficiency and high export volumes, importing sugar from abroad continues. Some experts call such a government decision a political move (Common Economic Space with Belarus). Others believe that the small volume of imports is a market partnership. Another opinion: in Russia, sugar is made not only from sugar beets but also from raw cane, the raw material for which does not grow in our country. Beets are harvested from late summer to mid-autumn. Furthermore, it is processed in factories into sugar. Even though this production method is the main one, this production may theoretically not be enough for the whole year. Then, there will be an acute sugar

deficit in the country. People will start to buy goods in bulk, and this can cause an increase in prices for it. The general excitement creates popular discontent, which the state will remove by artificially lowering costs. Therefore, there are alternative technologies for the production of sugar. In particular, the cane base is also used, which Russia imports. In any case, the trade surplus for this product cannot explain either the slowly decreasing trend in domestic rail transport of sugar or the change in its pricing policy.

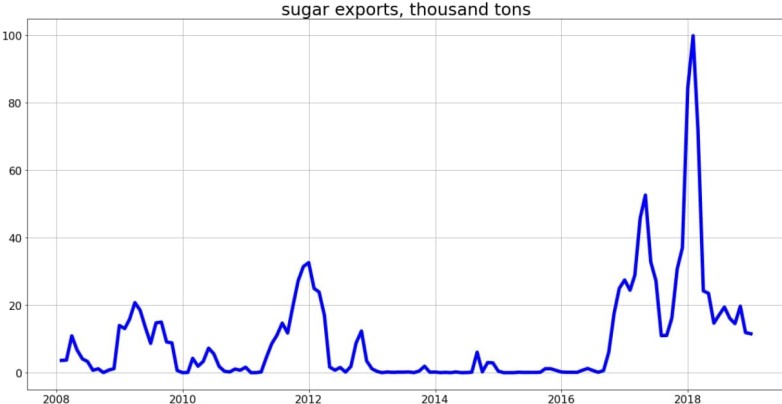

**Figure 17.** Export of sugar from Russia.

The next stage is the logistic redistribution of the transport load. Grain rail hoppers and particular purpose trucks mainly deliver sugar. Maritime transport also occupies a niche in the transportation sector; however, it is used for large-scale, long-distance deliveries. Trains have always been a priority for medium and long distances, while freight transport carried out delivery from the arrived train to the point of sale or vice versa. Therefore, one cannot complain about the logistical decision to replace some container transportations by rail with trucks. Under the main changes in the agricultural sector related to sugar, it is necessary to understand the dynamics in sugar beet yields (Figure 18) and the regulation of the volume of cultivated areas.

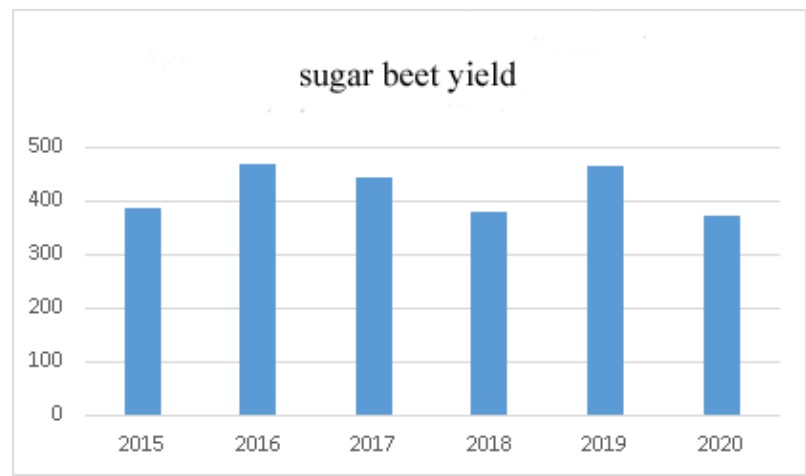

**Figure 18.** Productivity of sugar beet c/ha harvested area in the Russian Federation.

Producers choose favorable regions of the country for sowing and growing root crops. This is mainly the southern part of Russia, particularly the Krasnodar Territory. The indicator is quite unstable throughout the entire period of time, but there were no unproductive years for the period under consideration. In 2020, the country's southern regions experienced unfavorable weather conditions: the summer was dry, and the spring was cold. An active reduction in acreage accompanied the climatic situation. As a result, farms had a high proportion of re-seeding. As a result, the yield decreased by 23.00%,

and the average yield indicator in the period under review was 420 c/ha. This parameter cannot be called high, since the advanced economically developed countries have a similar indicator of more than 500 c/ha. However, until 2020, Russia managed to successfully supply the domestic market and supply the surplus for export. With not the highest yield, this can only be conducted with very large sown areas (Figure 19).

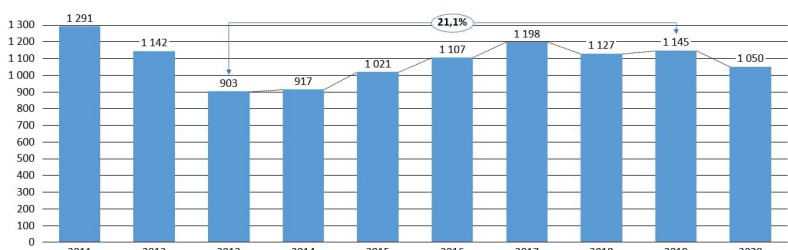

**Figure 19.** The sown area of sugar beet in the Russian Federation—a thousand hectares.

The chart shows an increase of 21.00% from 2013 to 2019. The Ministry of Agriculture decided to reduce the cultivated area in 2020, as in the last time, the volume of sugar production exceeded the level of consumption. The changes in the agricultural sector turned out to be very serious; there was a decrease in the importance of sugar beets. This could be a possible reason for a potential shortage in the sugar market. However, this needs to be verified by comparing consumption and production levels. When analyzing production, it is critical to say about the technological equipment of factories in the country. At the moment, about 70 enterprises are operating in Russia. Of these, about a third are owned by individuals, which contributes to the development of competition. This gives rise to introducing the latest technologies in factories and the integration of domestic manufacturers into world trade. At the same time, a good climate for the inflow of investment into the industry is created directly through such initiatives. At the same time, sugar beet is not a profitable agricultural crop. Figure 20 shows the production and consumption of sugar in the Russian Federation from 2010 to 2019.

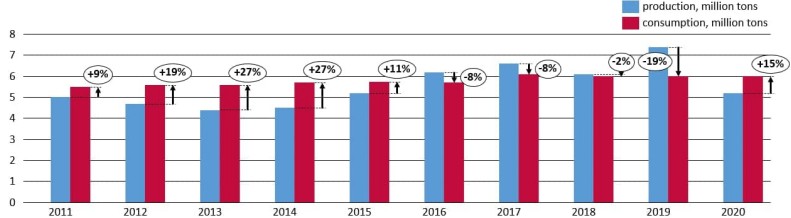

**Figure 20.** Production and consumption of sugar in the Russian Federation.

If in the early 2010s consumption exceeded production, then, starting in 2016, the situation begins to change dramatically. This can be explained by the fact that the country's total population in the period under consideration, according to Rosstat data, increased by more than one million people. The level of demand for sugar in 2019 remained approximately the same as it was in 2011. At the same time, the total tonnage of the sugar produced is gradually increasing. The production volume has dropped significantly in the last year, while consumption has not undergone significant changes. This dynamic is driven by the changes in the agricultural sector that were mentioned earlier. There was a deficit in the sugar market in 2020. Thus, in the Russian Federation, consumer price volatility can be explained by the population's expectation of a sugar shortage due to a decrease in crop yield and a difficult economic situation in the country and the world. Over the past five years, the average inflation rate has been relatively high—about 6.60%. This was accompanied by a decline in real incomes of the population in 2015, 2016, and 2017 by 3.20%, 6.00% and 2.00%, respectively. While the figure rose by 2.00% in 2018 and 2019, it was down by 3.50% again in 2020. All this testifies to a decrease in the purchasing power

of citizens, and, with positive population growth, the demand for sugar remains stable. In addition, the domestic rise in prices was supported by the global trend—in the 4th quarter of 2020, quotations for sugar futures increased significantly.

## 5. Conclusions

The paper proposed analyzing a comprehensive assessment of sugar prices in the Russian Federation in 2020. Transportation is an important component in determining the price. In open sources, there is no up-to-date data on the domestic transportation of the goods in question by rail, so we built the SARIMA predictive model. The direct support here was the indicators for the period from 2011 to 2019. As a result of the work of the model, a decrease in the volume of sugar transportation in 2020 was determined. Since the price of a product is inversely correlated with the number of goods transported, a reduction in supply should be accompanied by an increase in the price of sugar. Since the forecast contains a probabilistic nature, it must be substantiated empirically from an economic point of view. To do this, it was proposed to analyze the following factors: the volume of imports and exports of products, logistic redistribution of transport load, changes in the agricultural sector, the ratio of supply and demand for goods in the country. The situation on the sugar market in Russia is ambiguous. On the one hand, the Ministry of Agriculture is trying to prevent a crisis situation in this sector, finding the optimal yield and acreage ratio. On the other hand, 2020 has shown that the desire to cover domestic consumption does not always bring the desired result. The weather conditions greatly influenced the amount of the harvested crop. As a result, the volume of sugar production decreased, which caused a deficit, supported by the unfavorable situation in the country and the low purchasing power of the population. As a result, there was an active rise in prices, which was predicted using the SARIMA class model. This confirms the adequacy of our chosen methodology. Many other agricultural sectors can be assessed in the same way—with the right approach, planning accuracy will be high. The use of the predictive model considered in the article will make it possible to influence the increase in sustainable development of agriculture in any, even the most difficult, situations. This study has important insights for sustainable agricultural development. The results have important practical value and can be used to predict significant factors such as product price, sales volumes, imports and exports of agricultural products, etc. Those wishing to be acquainted with the full version of the study, data set, program code can send a request to the authors of this article.

**Author Contributions:** Conceptualization, S.K. and P.N.; formal analysis, D.S.; methodology, S.K. and R.G.; project administration, M.T. and P.N.; resources, I.B.; software, M.T. All authors have read and agreed to the published version of the manuscript.

**Funding:** This research received no external funding.

**Conflicts of Interest:** The authors declare no conflict of interest.

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
