# Peer review of "Analysis of the Forecast Price as a Factor of Sustainable Development of Agriculture"

_agronomy, doi:10.3390/agronomy11061235_

Round 1

Reviewer 1 Report

This paper's  title says price prediction, but the predictive variables used in the prediction model are not properly described. The predictive model also used the ARIMA model, but the model selection process doesn't seem very convincing. A lengthy description of the general content of the ARIMA model in the methodology is considered unnecessary in peer-reviewed papers. Specially, although the predictive power of the predictive model was visualized and presented, the numerical data for the error such as MSE from the measured value are missing.

There are several sentences where English expressions are also awkward, so it seems necessary to review English writing.

The method of citing references was not followed properly.

Reviewer 2 Report

A well scientifically documentated paper, with a very interesting case study, on the sensitive economic issues in agriculture and especially on the price of specific agricultural products, in the covid pandemia, such as sugar.

The technics, methods and models that are used are up-to dated (Learning Machine, Dickey - Fuller Criterion, Python Language).

The references and the literature, cover quite enough but need an enrichment. Some of the oldest scientific subjects that have been simulated and forecasted with ARIMA, S-ARIMA and other stochastic models and are related with the agicultural productivity are: Water Resources, Hydrology and Water Quality. So, could you please enrich the references, so that the paper to have a more global point of view, concerning such applications:  

Hongyan Du, Zhihua Zhao, Huifeng Xue, 2020. ARIMA-M: A New Model for Daily Water Consumption Prediction Based on the Autoregressive Integrated Moving Average Model and the Markov Chain Error Correction. Water, 12, 760, doi:10.3390/w12030760. 

Sentas A., Psilovikos A., Karamoutsou L.,  Charizopoulos N., 2018. Monitoring, modeling, and assessment of water quality and quantity in River Pinios, using ARIMA models. Desalination and Water Treatment, 133, (2018), 336–347, doi:10.5004/dwt.2018.23239.

Sentas A., Psilovikos A & Psilovikos T., 2016. Statistical analysis and assessment of water quality parameters in Pagoneri, river Nestos. European Water, 55, 115-124. 

Thi-Thu-Hong Phan, Xuan Hoai Nguyen, 2020. Combining Statistical Machine Learning Models with ARIMA for water level forecasting: The case of the Red River. Advances in Water Resources, 142, August 2020, 103656. 

Thank you!!!
